# An Enterprise Service Demand Classification Method Based on One-Dimensional Convolutional Neural Network with Cross-Entropy Loss and Enterprise Portrait

**DOI:** 10.3390/e25081211

**Published:** 2023-08-14

**Authors:** Haixia Zhou, Jindong Chen

**Affiliations:** 1School of Economics & Management, Beijing Information Science & Technology University, Beijing 100192, China; haixia0805@163.com; 2Beijing International Science and Technology Cooperation Base of Intelligent Decision and Big Data Application, Beijing 100192, China

**Keywords:** enterprise portrait, quality-service demand classification, 1D-CNN, cross-entropy loss

## Abstract

To address the diverse needs of enterprise users and the cold-start issue of recommendation system, this paper proposes a quality-service demand classification method—*1D-CNN-CrossEntorpyLoss*, based on cross-entropy loss and one-dimensional convolutional neural network (1D-CNN) with the comprehensive enterprise quality portrait labels. The main idea of 1D-CNN-CrossEntorpyLoss is to use cross-entropy to minimize the loss of 1D-CNN model and enhance the performance of the enterprise quality-service demand classification. The transaction data of the enterprise quality-service platform are selected as the data source. Finally, the performance of 1D-CNN-CrossEntorpyLoss is compared with XGBoost, SVM, and logistic regression models. From the experimental results, it can be found that 1D-CNN-CrossEntorpyLoss has the best classification results with an accuracy of 72.44%. In addition, compared to the results without the enterprise-quality portrait, the enterprise-quality portrait improves the accuracy and recall of 1D-CNN-CrossEntorpyLoss model. It is also verified that the enterprise-quality portrait can further improve the classification ability of enterprise quality-service demand, and 1D-CNN-CrossEntorpyLoss is better than other classification methods, which can improve the precision service of the comprehensive quality service platform for MSMEs.

## 1. Introduction

Insensitivity to market information and untimely access to information can easily lead to serious business risks for enterprises [1]. The enterprise comprehensive quality-service platform aims to help enterprises solve these problems. It provides *quality-service* products for MSMEs to improve the overall quality of enterprises. However, the normal enterprise service platform adopts the self-service mode. With a wide range of quality services and large differences, the problem of “information overload” has become increasingly prominent. Enterprises are not clear about their own quality status and do not timely obtain market information, resulting in inaccurate and incomplete quality-service products purchased on the platform. Therefore, their service efficiency is low, and the service is imprecise [2]. In e-commerce platforms that serve as third-party institutions, recommendation systems are an important tool to solve the problem of users’ inaccurate or incomplete search for platform service resources. Therefore, it is crucial to establish a service resource recommendation mechanism on the enterprise comprehensive quality-service platform to achieve efficient and accurate services.

In the recommendation system, the accurate acquisition of user preferences is the key to a personalized recommendation, which determines the effectiveness of the recommendation. Traditional recommendation algorithms use a “user-item” matrix for product recommendation, while enterprise service platforms have less enterprise user behaviors, sparse rating data, and cold start of the system [3]. According to relevant research, user profiles can help recommendation systems mine user behaviors and preferences, and expand user features to reduce data sparsity [4]. Based on the existing research ideas, this paper applied enterprise user profiling as a tool to enhance the recommendation effect. In addition, enterprise users have more attribute backgrounds than traditional individual users. Meanwhile, the preference for quality-service products of enterprises is closely related to the current quality situation of the enterprise operation. Therefore, enterprise quality features can be extracted from enterprise portraits for quality-service demand analysis. The types of service products purchased by enterprise users can be seen as their quality-service demands. The combination of enterprise portrait and service products is an effective path to improve the platform’s precise service level.

Enterprise portrait is an important method to portray the characteristics of enterprises, and includes enterprise credit [5], enterprise finance, enterprise taxation [6], enterprise business development and risk identification [7], etc. Different dimensions can reflect different aspects of the enterprise quality status. Quality-service, like other products, requires service providers to have a deep understanding of service demands and strengthen information interaction [8]. The enterprise comprehensive quality portrait portrays the multidimensional quality condition of an enterprise, which includes not only the quality of products but also the quality of management, operation, and innovation.

In summary, the purpose of this paper is to introduce the enterprise portrait for enterprise users, extract quality characteristics of enterprises, and integrate enterprise comprehensive quality portrait labels to establish a classification method for enterprise quality-service demands. The loss function is the difference between the predicted value of the model and the true value for a specific sample. This paper applies cross-entropy loss to optimize the service demand classification model. The main content of this paper includes the following: First, the enterprise portrait concept is used to realize the portrayal of enterprise quality information, and the enterprise data are processed to obtain the enterprise quality characteristics labels; second, the service transaction information of the enterprise service platform is used to analyze the enterprise quality-service demand; then, the portrait labels are used to concatenate with the service categories in the transaction, and the service categories selected by the enterprise are used as the quality-service demand labels to form a sample of supervised classification methods; finally, we use cross-entropy loss to optimize our service demand classification model. Based on the classification results of the demands, the relevant quality-service products can be recommended to enterprise users.

## 2. Related Research

### 2.1. User Demand Mining in Recommendation Systems

With the rise of artificial intelligence, a large number of machine-learning and deep-learning methods have been applied to user interests mining research. Exploring users’ demands and preferences is a hot issue in recommendation system research. The main recommendation algorithms include content-based recommendation, collaborative filtering recommendation, association rule-based recommendation, and hybrid recommendation.

The basic idea of content-based recommendation methods is to construct recommendation models by tagging user and product/service information, for example, libraries combine reader information and book information [9]. Collaborative filtering recommendation looks at the user’s purchase history, product reviews, and product labels to calculate the product feature matrix similarity and calculates the user feature matrix similarity based on the user’s historical behavior characteristics to recommend items for the user [10]. Association rule-based recommendation [11] is the conclusion of different rules derived from data analysis to represent the implicit association that exists within the data, i.e., to find the dependency or correlation between events and events. Hybrid recommendation algorithms usually use a combination of multiple recommendation methods to compensate for the deficiencies between the different methods to obtain a better recommendation solution [12].

However, compared with individual users, enterprise users lack user browsing behavior and rating data, which makes the “user-item” rating matrix of the recommendation system of the enterprise service platform difficult to obtain. Enterprise users have more attribute backgrounds, and the enterprise portrait can improve the efficiency of the recommendation system by extracting labels for management, innovation, business risks, and other backgrounds and classifying quality-service requirements.

### 2.2. Portrait in Recommendation

In recent years, portrait technology has been widely used in enterprise precise services and personalized recommendations. The core basis of current common recommendation algorithms is the acquisition of user preferences [13]. Therefore, many scholars use portraits to predict behavioral preferences and service demands. For example, Zhang Y. et al. used user dynamic portraits to monitor the feedback of user behavioral dynamics in real time and performed KNN classification modeling to predict user demand and behavioral preferences [13]. 

The research of personalized recommendation using portrait labels has been widely used in book lending, e-commerce platforms, social media, health care, and other fields, and also can play a technical guidance role for accurate services in other industries and fields. Huang J. et al. [2] established an enterprise portrait containing 16 characteristic elements of enterprises to realize the precise service of enterprise industry information, which is inscribed in the industry information precision service of enterprise portrait. Li X. used the two-way portrait of enterprises and science and technology service products to realize the precise service research of enterprise science and technology service platforms [14]. Song K. et al. conducted enterprise portraits from two perspectives of enterprise technology R&D attributes and competitor attributes to realize patent recommendations between schools and enterprises [15]. 

The precise recommendation based on enterprise portrait have achieved preliminary results, but the portrait dimension is still inclined to the extraction of labels for specific field problems, such as the application to the portrait of patents, the level of electricity consumption and financial financing. However, it lacks the portrait of the comprehensive quality of enterprises. This paper combines the comprehensive quality-service products to study the precise service and personalized recommendation of the quality-service platform.

The service recommendation based on enterprise portrait is a hybrid recommendation approach combining content-based and collaborative filtering. Related research commonly uses Bayesian networks [16] for user similarity calculation, XGBoost for enterprise classification prediction [17], and Apriori algorithms for label–demand–behavior association analysis [18]. There are studies to solve the problems of computation and recommendation accuracy caused by the missing data rate and sparsity by plain Bayesian [19] and KNN-SVM [20], etc. Feng Z. enhanced sparse data sets based on deep neural networks and optimized similarity metrics to improve the recommendation accuracy of recommendation algorithms [21].

In the classification of quality-service demand for enterprises, more accurate classification results are expected. Comparing the output results of the model with actual data can measure the performance of the model. Cross-entropy can easily calculate the relationship between them. Adjusting the model through cross-entropy loss can optimize the model, so as to better help the platform recommend service products to enterprise users [22].

Based on the above research exploration of portrait-based service recommendation, this paper establishes the enterprise service demand classification method. The service demand classification model incorporating enterprise portraits demands to first perform feature extraction of enterprise users through enterprise portraits, excavate potential preferences of enterprise users, and gradually optimize the model through cross-entropy loss function to complete the enterprise quality service demand classification.

## 3. Methodology

According to the definition of quality technical services in the national standard [23], the comprehensive quality services for enterprises can be understood as technical services in measurement, standards, inspection, testing, and certification and accreditation for enterprise products, services, personnel, institutions, or other objects. Quality services involve complex and diverse service products. Through enterprise quality characteristics the current situation of enterprise comprehensive quality is analyzed. There are important methods for the platform for comprehensive quality service to carry out accurate services.

As shown in Figure 1, first, when registering an account, MSME users provide their own structured characteristic attributes *D_i_* (such as company industry type, company attribution, company type, etc.), and the platform can build a user portrait model to obtain an enterprise quality label *L_i_*; Second, when enterprise users purchase quality-service products *q_m_* on the platform, a transaction behavior occurs. Then, the platform stores the transaction relationship of “enterprise-service”. Based on this transaction relationship, *q_m_* can be regarded as the enterprise’s demand for quality-service products. We use the category *Q_j_* of quality-service products *q_m_* as the true value *y* for the service demand classification. Finally, the experimental dataset {*L_i_*, *Q_j_*} for MSME is formed. 1D-CNN-CrossEntorpyLoss mines predicts the quality service demand *Q_j_* of enterprise users. According to the category *Q_j_* output by the model, the recommendation system can recommend products of the same category to users.

### 3.1. Enterprise Quality Portrait Label

In this paper, we initially establish a portrait label dimension containing industrial and commercial attributes: management quality, product quality, industry qualification, innovation quality, and operation quality based on the concepts of the comprehensive quality services.

Management quality is the outstanding performance of the enterprise in terms of quality management systems and enterprise quality objectives. Liu Ying pointed out that many manufacturing enterprises need to have clear quality objectives, quality concepts, and quality culture aspirations in quality management; product quality is the enterprise quality that can be directly perceived by the market and consumers, including both the product itself and external perception [22]. Industry qualification is a necessary qualification for the enterprise to carry out production and operation, and it is a representative of the enterprise’s long-term capability and a factor that can represent whether enterprises can develop stably in the long run; innovation quality reflects the enterprises’ innovation consciousness as well as the output capacity of innovation results. Operation quality judges whether enterprises can operate and develop smoothly through their operation activities and the risk level in judicial activities. Later, according to the labeling system, we can collect the relevant index data from the comprehensive quality-service platform, *GongSiBao*, government websites, etc., and clean the statistics for enterprise portrait label extraction.

### 3.2. “Label-Service” Vector Concatenating

There exists a transaction relationship {*u_k_*,*q_m_*} between enterprise user *U* and quality service *Q*. Enterprise user *u_k_* also has quality attribute information and portrait label characteristics. The enterprise portrait data are represented as *U* = {*L*_1_, *L*_2_, *L*_3_, …… *L_n_*}, where the enterprise quality label of each dimension *Li = {l_i_*,*a_i_*_1_, *a_i_*_2_, *…… a_iw_}* contains the comprehensive quality information *a_ij_* of the enterprise. The method of vector concatenation is shown in Figure 2. The transaction relationship {*u_k_*,*q_m_*} between enterprise and service and the enterprise’s own characteristics *L_i_* will be connected through the enterprise *ID.* Then, we will be able to obtain the enterprise’s “label-service” sequence data {*u_k_*,*L_i_*,*Q_j_*}, which represent the service preferences of each enterprise. These data will be fed into the convolutional neural network for deep mining potential enterprise-service relationship. So as to achieve the effect of quality-service demand prediction and quality-service recommendation.

### 3.3. Cross-Entropy Loss

The concept of entropy, which was proposed by the German physicist Clausius in 1877, is a function of the state of the system, where a reference value and the variation in entropy are often analyzed and compared. Cross-entropy (CE) is a type of entropy that reflects the similarity between variables from the perspective of probability [24]. Enterprise quality-service demand classification is a supervised classification task, and the sample labels for supervised training have been determined when model training is performed, so the true probability distribution is shown in Equation (1):(1)$$H\left(X\right)=-\sum_{i=1}^{n}p(x_{i})log(p(x_{i}))$$

Cross-entropy measures the degree of difference between two different probability distributions in the same random variable and is expressed in machine learning as the difference between the true probability distribution and the predicted probability distribution [25]. Cross-entropy is used in combination with *Softmax* to process the output, such that the sum of the probabilities of multiple classification predictions is 1. Cross-entropy is then used to calculate the loss, and the smaller the value of cross-entropy, the better the performance of the classification model [26].

Therefore, the model’s cross-entropy loss function (Cross-Entropy Loss) is defined in the single-label classification task assuming a true distribution of *y*, a network output distribution of $\hat{y}$, and a total number of categories of *n*. The cross-entropy loss function is shown in Equation (2):(2)$$CrossEntorpyLoss=-\sum_{i=1}^{n}y_{i}log(\hat{y_{i}})$$

### 3.4. 1D-CNN

The basic structure of a convolutional neural network (CNN) consists of an input layer, a convolutional layer, a pooling layer, a fully connected layer, and an output layer. The convolutional layer and the pooling layer can be set alternately. The convolutional layer consists of multiple feature maps, and each feature map consists of multiple neurons, each of which is connected to a local region of the previous layer through a convolutional kernel [26]. In this study, the sequences of enterprise quality labels are stitched with quality-service transaction behaviors, the sequence data of quality features are input, and one-dimensional convolutional neural network (1D-CNN) is used for enterprise quality-service demand prediction. A schematic diagram of the 1D-CNN structure is shown in Figure 3.

The 1D-CNN model has five main layers as follows:

(1) Input layer: Quality labels *L_i_* (*i* = 1, 2, 3, ……, *n*) from the enterprise comprehensive quality portrait are used as the input data, and the type of quality service *Q_j_* is used as the output result of the model.

(2) Convolution layer: The data *L_i_* (*i* = 1, 2, 3, ……, *n*) are fed into the convolution layer using a sliding window for convolution operation to obtain the enterprise user quality feature values, and the calculation formula is as follows:(3)$$c_{i}=f(w_{e}\cdot l_{i:i+h-1}+b)$$
where $l_{i:i+h-1}$ denotes the *h* data adjacent from the *i*th position as a sliding window; *f* denotes the activation function; $w_{e}$ denotes the convolution kernel size; $c_{i}$ denotes the feature value at the *i*th position; and *b* denotes the bias.

(3) Pooling layer: In this paper, the maximum pooling operation is used to change the length of the labeled data and obtain the most important features in the Feature Map (Feature Map).
(4)$$\overset{\wedge }{c}=max\{c\}$$

The maximum eigenvalues of all convolutional layers are then obtained to generate the high-level eigenvectors of the data:(5)$$V=[{\overset{\wedge }{c}}_{1},\;{\overset{\wedge }{c}}_{2},\cdots ,\;{\overset{\wedge }{c}}_{m}]$$
where *m* is the number of convolutional kernels.

(4) Fully connected layer: The feature vector *V* is flattened and then input to the fully connected layer, which can integrate the local information with category differentiation in the convolutional or pooling layers.

(5) Output layer: The output of the final fully connected layer is fed into the output layer, which can be classified using *Softmax* to determine the service requirements of each enterprise sample.
(6)$$y=Softmax(w_{s}v+b_{s})$$
where $w_{s}$ denotes the weight, *s* denotes the number of categories, and $b_{s}$ denotes the bias.

### 3.5. Experimental Steps for 1D-CNN-CrossEntorpyLoss

After the enterprise portrait processes the enterprise quality data, the enterprise quality label sequence is obtained and fed into 1D-CNN-CrossEntorpyLoss for training, and the specific experimental steps are shown in Figure 4.

The model training process has the following four steps:

**Step 1**: According to Section 3.2, the enterprise label and service type *Q_j_* are concatenated according to the enterprise ID to form “label service” sequence data. The enterprise quality labels under each service category *Q_j_* are obtained.

**Step 2**: The data obtained from **Step 1** are divided into a training set and a test set according to 4:1.

**Step 3**: The training set is used to extract the features of the enterprise quality labels by 1D-CNN. The model will classify them with the *Softmax* function, calculate the cross-entorpy loss, and optimize the network parameters step-by-step with the Adam optimization [27] algorithm to obtain the 1D-CNN-CrossEntorpyLoss quality demand classification model.

**Step 4**: Test samples are fed into the trained 1D-CNN-CrossEntorpyLoss classification model. The model will output the quality-service classification results, which will be used as the basis for the platform and service providers to analyze the enterprise service demand.

## 4. Experiments and Discussions

### 4.1. Enterprise Portrait Dataset

#### 4.1.1. Portrait Extraction

In Figure 5, according to the dimension of the enterprise portrait, 35,439 samples’ enterprise data are collected. The information on the indicators involved is obtained from multiple channels such as *GongSiBao*
https://www.gongsibao.com/ (accessed on 9 August 2023), quality-service platform. Data cleaning and pre-processing are performed on the data. There are 35,369 enterprise samples. The data has removed incomplete samples and samples of enterprises that have already been deregistered. 

According to different application scenarios and label attributes, different label extraction methods can be used for label setting: (1) Business and industry attribute class indicators, such as enterprise size, industry type, registered capital, etc., can be directly used as factual labels; (2) For indicators such as product pass rate, sampling pass rate, etc. can be based on business knowledge practice and understanding of industry, business, scenarios, and problems for quality analysis, so the threshold method can be used to rule; (3) For text data such as business scope and penalty reasons, keywords extraction and text summary can be performed to show the characteristics of the category; (4) For comprehensive labels of innovation capability category, data standardization can be performed, after which data reduction, indicator screening, cluster analysis, and classification can be used to represent numerous attribute data as specific and unique category labels. The enterprise quality labels are shown in Table 1.

#### 4.1.2. Data Processing

Enterprise quality features contain continuous features and discrete features (category features). Continuous features require normalization. Discrete features are processed in the following two ways. (1) *One-Hot Encoding*: character features are transformed into [0, 1] numerical features with *one-hot encoding*; *one-hot encoding* hosts each state of a feature separately, that is, a feature has *n* taken values, the feature will be extended; The *one-hot encoding* can extend the value of discrete features to the Euclidean space, which is more reasonable when calculating the feature distance. (2) *Feature mapping*: the factorize function of python’s panda’s library (referred to as “*pd. factorize*”) of the panda’s library in python to transcode the values of character features to complete feature mapping, which shows convenience when dealing with features with more values. When a feature has more values, *one-hot encoding* can cause dimensional disaster and makes the computation surge, at this time, using *pd.factorize* can keep the feature as 1 column and reduce the computation; when the feature has only two values, the feature can be changed to {0,1} values, reducing the computation while reducing the dimension.

### 4.2. Parameter Setting and Model Structure

In Table 2, 2668 orders are collected from the quality-service platform *GongSiBao*
https://www.gongsibao.com/ (accessed on 9 August 2023) of the information technology service industry. Two types of service product transaction records of “intellectual property” and “value-added telecommunications” are used as the experimental data to build the corresponding quality-service demand prediction model.

The experiments are performed on PC with a 64-bit Windows 10 operating system. The model is optimized using the Adam = algorithm and cross-entropy loss. The learning rate is set to 0.001, and the size of the convolution kernel is 3; the size of the pooling kernel of max-pooling pooling layer is 2; the dimension is 2, and the *Softmax* output is set to 2. During the training process, each sample in the training set is processed, the output results are compared with the label classification results, and the models are evaluated using the test set evaluation, comparing the accuracy, precision, recall, and AUC performance metrics of each model.

### 4.3. Experimental Results

#### 4.3.1. Data Feature Processing

To improve the model effect, *one-hot* encoding using *pd.get_dummies* and feature mapping using *pd.factorize* are used to compare the model data feature processing methods and choose the processing method with the best model performance. The model performance under different feature processing methods is shown in Figure 6. The overall performance of the model is optimal when *one-hot* is performed on the enterprise quality label data, so this paper uses *one-hot* to process the data features.

#### 4.3.2. Model Parameters Selection

To investigate the effects of learning rate [18] and Dropout [28] on the model performance, multiple sets of experiments were conducted on the training set. The model accuracies for different learning rates and different Dropouts are shown in Figure 7. The effect of the learning rate on the model accuracy of the validation dataset is shown in Figure 7a. The experimental results show that the performance of MSMEs quality-service demand prediction model is optimal when the learning rate is 0.001. Dropout temporarily discards the neurons in the network according to the set probability so that the discarded neurons do not participate in training. Dropout achieves the purpose of simplifying the model and can have good classification performance in the validation set. Figure 7b shows the effect of different Dropout values on the model accuracy, and Dropout of 0.25 is chosen for model construction.

#### 4.3.3. Comparison of Iterative Process of 1D-CNN-CrossEntorpyLoss on Training Set and Validation Set

As can be observed in Figure 8, the 1D-CNN-CrossEntorpyLoss model is trained 50 iterations, in which the cross-entropy loss on training set and test set gradually decreased. The cross-entropy loss on the training set was smaller than that on the test set, but the cross-entropy loss on the test set decreased to approximately 0.575, which would fluctuate slightly after the 5th iteration and finally gradually converged to a stable state.

Through the experimental comparison of universal classification models, it can be found that the final prediction accuracy of 1D-CNN-CrossEntorpyLoss and the mean values of each evaluation index show that the model is better than other benchmark models. The experiment verifies the practicality of the comprehensive quality portrait of enterprises for providing accurate quality services. The model performance indicators are taken as the mean values of various types of service prediction indicators as shown in Table 3 below.

1D-CNN service demand prediction model is a classification model based on enterprise quality labels and service transactions, which can provide data basis and decision support for analyzing enterprise quality demand by extracting enterprise quality labels. To verify the effectiveness of extracting quality labels and constructing enterprise portraits in this paper and to improve the effectiveness of the quality-service demand prediction model, data ablation experiments of 1D-CNN-CrossEntorpyLoss model are conducted on labeled enterprise quality labels {*L*}, unlabeled quality data {*D*}. It can verify the effects of dataset {*L*}, dataset {*D*} and dataset {*L, D*} on model performance, and to determine the effect of enterprise quality labels {*L*) on the positive effect of enterprise quality-service demand analysis.

As shown in Table 4, by comparing the model effects in different data set cases, it is found that the demand prediction results based on the original enterprise quality data {*D*} have the lowest accuracy, the service demand prediction using enterprise quality labels {*L*} has higher accuracy, and the service demand prediction effect is the best after adding enterprise quality labels {*L*} and original data {*D*}. The experimental results show that the quality labels extracted by enterprise portraits can extract important features about comprehensive quality services. The enterprise portraits constructed in this paper and the label extraction method can effectively extract enterprise quality features, analyze enterprise quality service demands and provide decision support to the service platform.

Comparing the predicted results with actual transaction data, we found that the accuracy can reach over 70%. That is to say, when we recommend products based on the predicted results, more than 70% of users will purchase the recommended products. Therefore, we believe that 1D-CNN-CrossEntorpyLoss can improve the quality of service recommendations. It is effective to use the results of requirement classification as a decision for recommendation systems. This study further verifies the ability of cross-entropy loss to improve the performance of convolutional neural network models and the effectiveness of the comprehensive quality-service-oriented MSME portrait constructed in this paper to improve the level of accurate service.

## 5. Conclusions

Starting from the concept of comprehensive quality, this paper constructs the comprehensive quality portrait of enterprises for MSMEs on the comprehensive quality-service platform. The types of service products purchased by enterprise users are considered as their quality-service demands. Through feature extraction of enterprise quality labels and historical transaction records, the comprehensive quality-service demand classification of MSMEs is realized, and the service efficiency is improved for the platform. The empirical study finds that for discrete enterprise quality labels, one-hot coding of data features can improve the accuracy of model classification. Compared with the original quality data, the quality labels extracted with the portrait model can significantly improve the accuracy of quality-service recommendations. The portrait labels can accurately describe user characteristics, structure enterprise information, and improve the accuracy of service demand classification.

The comprehensive quality portrait of enterprises can accurately describe user characteristics, structure enterprise information. By combining the demands with quality-service products, enterprise portrait can accurately identify the quality-service needs of enterprise users, and provide a basis for precise marketing and recommendation of quality-service platforms. In the future, the study can still be carried out in the following areas: (1) The quality profile of enterprises can not only be analyzed through public data, but also be enriched through on-site research to provide more accurate feature labels. (2) Secondly, there may be differences in the impact of different portrait dimensions and labels on demands, and the contribution of each attribute *X* to the results is different. Therefore, this can be considered to adjust the 1D-CNN-CrossEntorpyLoss and divide it into more fine-grained requirement types for more accurate recommendations. (3) Finally, in the process of building a recommendation model based on the classification results of enterprise users’ needs, the feature profiles of enterprise users and quality services can be bidirectional matched to improve service recommendation effectiveness.

## Figures and Tables

**Figure 1 entropy-25-01211-f001:**
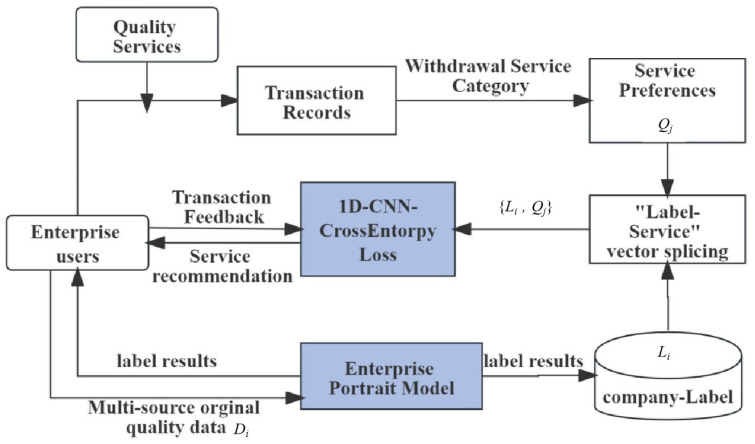
Enterprise Service Demand Classification Method based on 1D-CNN-CrossEntorpyLoss.

**Figure 2 entropy-25-01211-f002:**
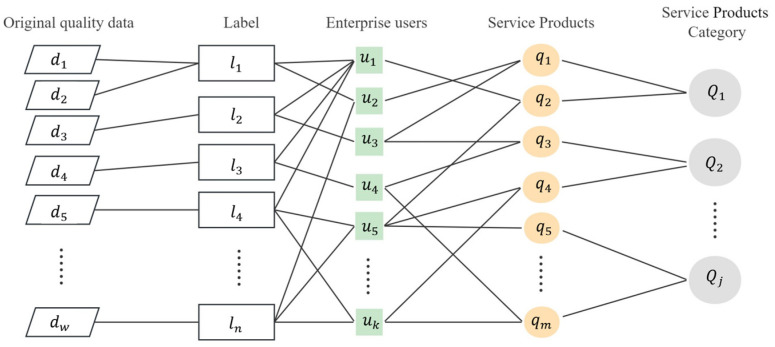
The concatenation method of enterprise quality label and quality services.

**Figure 3 entropy-25-01211-f003:**
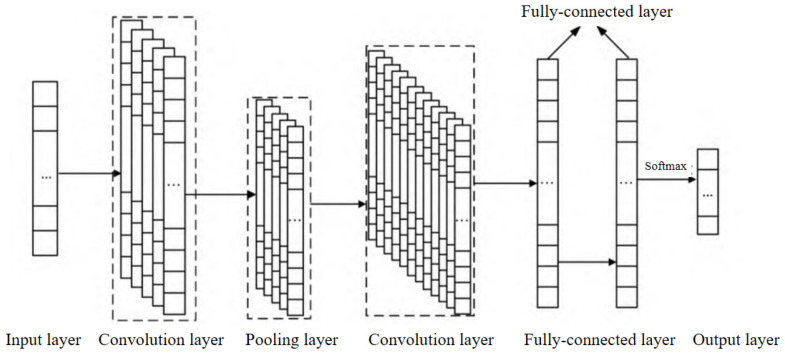
The structure of a one-dimensional convolutional neural network.

**Figure 4 entropy-25-01211-f004:**
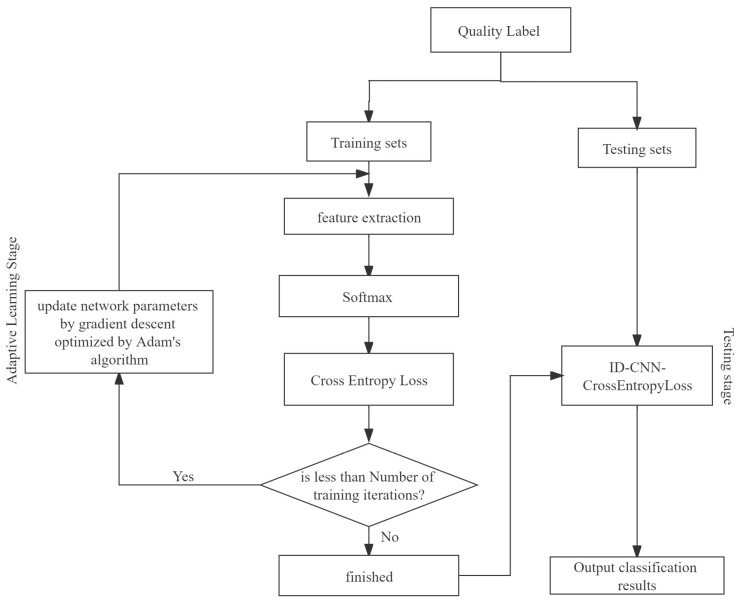
The specific experimental steps of 1D-CNN-CrossEntorpyLoss.

**Figure 5 entropy-25-01211-f005:**
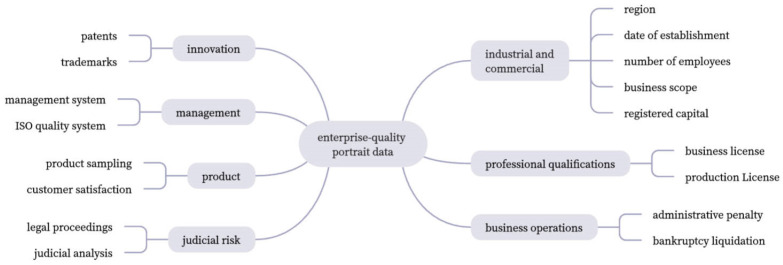
Data for the enterprise portrait from *GongSiBao.*

**Figure 6 entropy-25-01211-f006:**
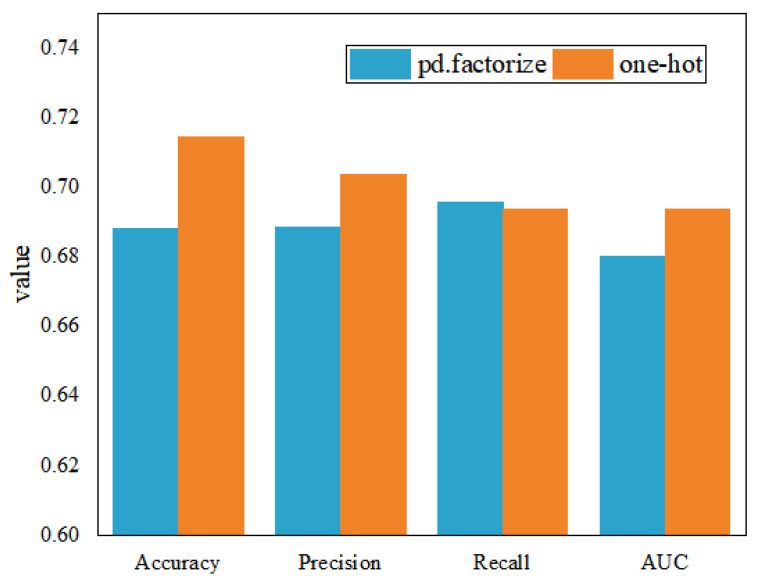
Effect of different feature processing methods on model training set results.

**Figure 7 entropy-25-01211-f007:**
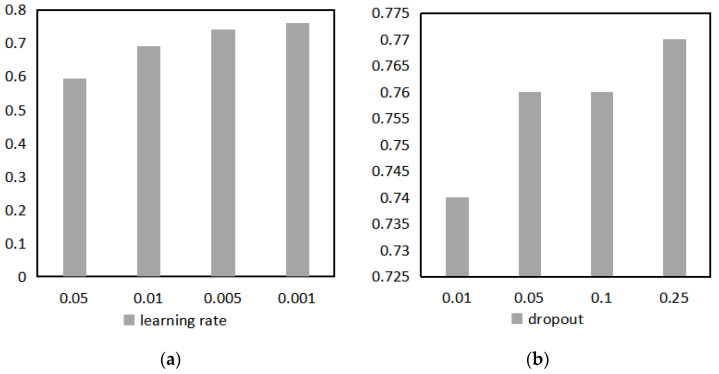
Effect of learning rate and Dropout on the accuracy of the model on the training set: (**a**) Learning rate impact on model performance; (**b**) Dropout impact on model performance.

**Figure 8 entropy-25-01211-f008:**
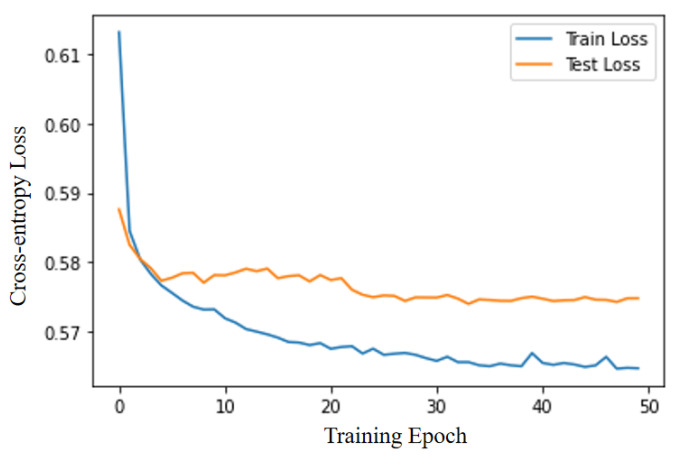
1D-CNN-CrossEntorpyLoss training process.

**Table 1 entropy-25-01211-t001:** Enterprise portrait label data in quality-service platform (partial).

Company ID	Region (*L*_1_)	Industry Experience (*L*_2_)	Enterprise Size (*L*_3_)	Innovation Level (*L*_4_)	Operational Risk (*L*_5_)	Judicial Risk (*L*_6_)
43659737	Beijing	Rich	Medium	Ⅲ	Ⅰ	Ⅰ
34894090	Shanghai	General	Small	Ⅱ	Ⅰ	Ⅰ
5151699	Tianjin	Less	Micro	Ⅰ	Ⅱ	Ⅱ
44167631	Xi’an	Average	Small	Ⅱ	Ⅲ	Ⅲ
27164432	Beijing	average	Small	Ⅱ	Ⅵ	Ⅵ
130251713	Guangzhou	Less	Micro	Ⅱ	Ⅰ	Ⅰ
41824100	Shijiazhuang	General	Medium	Ⅱ	Ⅰ	Ⅰ
67654723	Beijing	Rich	Medium	Ⅵ	Ⅰ	Ⅰ
43659737	Tianjin	Less	Small	Ⅰ	Ⅲ	Ⅲ
34894090	Qingdao	Rich	Medium	Ⅵ	Ⅰ	Ⅰ

**Table 2 entropy-25-01211-t002:** Details of orders from the quality-service platform.

	Column	Comment
1	company_id	Enterprise Unique ID
2	order_id	Order Unique ID
3	produce_type_pkid	Type of service products purchased by the enterprise, *Q*
4	produce_price	price
5	order_price	Total order price
6	order_add_time	Time of order

**Table 3 entropy-25-01211-t003:** Comparative experimental results of comprehensive quality-service demand classification.

Model	Accuracy	Precision	Recall	AUC
Logistic	0.6872	0.6908	0.65	0.6372
SVM	0.7053	0.6813	0.6937	0.6684
XGboost	0.7025	0.7046	0.6808	0.6372
**1D-CNN-CrossEntorpyLoss**	**0.7244**	**0.7139**	**0.6938**	**0.6938**

**Table 4 entropy-25-01211-t004:** Classification effect of 1D-CNN-CrossEntorpyLoss under different datasets.

Dataset	Accuracy	Precision	Recall	AUC
{*D*}	0.4041	0.2021	0.5	0.5
{*L*}	0.7244	0.7139	0.6938	0.6938
**{*L*, *D*}**	**0.7491**	**0.7541**	**0.8665**	**0.7186**

## Data Availability

Data was obtained from the third part *GongSiBao* and are available at https://www.gongsibao.com/ with the permission of *GongSiBao*.

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
