# Peer review of "An Enterprise Service Demand Classification Method Based on One-Dimensional Convolutional Neural Network with Cross-Entropy Loss and Enterprise Portrait"

_entropy, 2023, doi:10.3390/e25081211_

Round 1

Reviewer 1 Report

The authors introduce an enterprise service demand classification method that uses Convolutional Neural Network which is trained using enterprise portrait data obtained from various systems together with enterprise quality labels and historical transaction records,

The main goal highlighted in the article, the recommendation system improvement, is important and topical, but the results do not reflect the improvement of the recommendation system. As a result, the article’s purpose as a whole is not clear in this context.

You can find the detailed remarks below:

1)      In the introduction section and related research section, authors suggest that the main purpose of the research is recommendation system improvement, however, they do not provide a convincing relationship between enterprise demand prediction and recommendation system. In general recommendation system is based on customer behavior, while the enterprise portrait concept uses company data. Can you provide reliable confirmation that such a relationship exists?

2)      The method bases on Chinese data acquisition systems. The reviewer has no access to it, and the results were not demonstrated on other datasets, so the results are difficult to verify. As a result, the method introduced in the article is not general and impossible to verify.

3)      In general MSMEs are flexible and may not provide much information to the external world (it may work differently in China, but the algorithm should be general). As a result, the publicly available information may be completely inadequate for what is going on in the MSME effectively. This means, that the enterprise portrait may be inadequate, which is very common in European Union, for example. Such an approach makes sense for large companies where more precise reports are available in specified formats. Can you provide examples of your algorithm implementation from different countries, together with some convincing proof of the reliability of such data for MSME in these countries?

4)      Convolutional neural network is not the best choice to use in the application which is introduced in the article, as in general, CNN is designed for large (mainly image-based) datasets. In this sense, the dataset which is used in this research is not large. Probably regular neural network would also produce similar or even better results. Can you justify why you are using CNN, not LSTM, for example, which proved its performance in time series prediction?

5)      According to the explanation in lines 153 and 155, there are Li and Di inputs, but they are not marked in Fig. 1

6)      It is unclear how the reference values are obtained. How reliable are the reference data? Do you predict the demand in time and verify I with real demand? How are you sure that this is the overall demand of the company? Does the prediction concern the company as a whole or is this the prediction for selected service? If you do so, how do you obtain the reference data for recommendation in such a case?

7)      In the conclusions section you say that you improve the quality of service recommendation while you actually predict (or, in fact, classify) service demand. You do not provide the data concerning recommendations.

8)      You write that: “The enterprise comprehensive quality portrait can analyze and model the target enterprise user group”. In which part of the article have you proved it? You should obtain this information from users not from quality portrait, or at least take the information from users into account, but it is unclear how do you take the users into account in your algorithm.

Reviewer 2 Report

The paper presents a quality service demand classification method (12 1D-CNN-CrossEntorpyLoss), based on Cross-Entropy Loss and one-dimensional convolutional neural network (1D-CNN) with integrated enterprise quality portrait labels. In this paper, the Authors applied Cross-Entropy to build a loss function to minimize the loss of the classification effect of the 1D-CNN model and enhance the performance of enterprise quality service demand classification. Moreover, the Authors compared their approach with XGBoost, SVM, and logistic regression models as well as provided experimental results. The topic is interesting and the paper well corresponds with the journal’s aim and scope.

However, there are shortcomings in this paper. In the Introduction, the purpose of the article is slightly underlined in the text - it is worth starting with a new paragraph. I suggest the Authors highlight strongly their contribution. The section 2 is complete. In Section 3, Figure 3 is not clearly depicted. In section 4.1 please add information about the origin/source of the data, similarly, in section 4.2. In the Conclusions section, future works should be added.

Overall, the paper looks good, but some parts need to be improved.  

Round 2

Reviewer 2 Report

As it was written in the 1st revision, the paper presents a quality service demand classification method (12 1D-CNN-CrossEntorpyLoss), based on Cross-Entropy Loss and one-dimensional convolutional neural network (1D-CNN) with integrated enterprise quality portrait labels. The Authors were requested to make some improvements to the paper, including adding a new paragraph to the Introduction section, enhancing Figure 3, providing missing information about the data, and including future works. The Authors have successfully addressed all of the comments.